# Deficits of Sensory Integration and Balance as Well as Scoliotic Changes in Young Schoolgirls

**DOI:** 10.3390/s23031172

**Published:** 2023-01-19

**Authors:** Jacek Wilczyński, Natalia Habik Tatarowska, Marta Mierzwa Molenda

**Affiliations:** Posturology Laboratory, Collegium Medicum, Jan Kochanowski University, Ul. Żeromskiego 5, 25-369 Kielce, Poland

**Keywords:** neurophysiological background of scoliosis, Diers Formetric III 4D optoelectronic method, Biodex Balance System platform

## Abstract

The aim of this study was to assess the relationship between sensory integration and balance deficits as well as scoliotic changes in young schoolgirls. The study comprised 54 girls aged 11 years with scoliotic changes. The Clinical Test of Sensory Integration and Balance of the Biodex Balance System platform were used to analyze the deficits in sensory integration and balance. Scoliotic changes were assessed using the Diers Formetric III 4D optoelectronic method. In the present study, there was a significant relationship between sensory integration and balance deficits as well as spine curvature angle (°) (*p* = 0.01), vertebral surface rotation (°) (*p* = 0.03), pelvic tilt (°) (*p* = 0.02), and lateral deviation (mm) (*p* = 0.04). The integration of the sensory systems has a positive effect on the structure of the intended and controlled movement as well as body posture and the development of the spine. In the treatment of scoliotic changes, one should also consider exercises that improve sensory integration as well as position and balance reactions.

## 1. Introduction

Sensory integration (SI) is the process by which the central nervous system (CNS) receives, segregates, recognizes and processes information in order to respond to it with a motor response. It is related to the organization and ordering of stimuli reaching the CNS, which come from various receptors. The integration of sensory systems affects the structure of intended and controlled movement as well as body posture. Due to such a specialized system of sensory integration, the functioning of motor coordination and the quality, speed, and fluidity of movement become improved [1,2,3]. Both the excess of information and the scarcity or inability to interpret the strength, extent, and direction of action result in various disturbances. Disturbances in the sense of balance and in deep and superficial sensations may be caused by hypofunction, i.e., reduced insufficient sensitivity of the senses or hyperfunction, i.e., excessive sensitivity [4]. The development of the body posture regulation system reflects the changes taking place in the CNS, mainly its maturation and functional improvement. It is synchronized with posturogenesis, as well as with the development of other elements of the anti-gravity system [5]. The stage of integrating flexion and extension into antigravity activity enables the development of co-contraction, i.e., the ability to maintain proper tension around the joints in the form of their external stabilization [6]. This makes it possible to generate sufficiently high values of muscle tension in segments distant from the head but located close to the central line of the body. Body mass is stabilized in further areas of the body, i.e., the pelvis and lower limbs, which provides the internal control of body position. In addition, the development of balance reactions, resulting from the maturity of the area around the cerebral cortex, allows it to achieve full control regarding the position of the body’s center of gravity, thus achieving so-called central control [7]. Full differentiation of motor activities takes place when the stability and mobility of all body segments is under the control of automatic balance reactions. The CNS controls body posture based on information from proprioceptors as well as the vestibular and visual systems. All these sensory systems are interconnected and integrated, creating a properly functioning anti-gravity system in the CNS. This system further allows for the harmonious and effective postural development of a child [8]. AI processes play a significant role in maintaining proper postural tension and postural stability, which influence the development of body and spinal posture. The task of the CNS is to organize sensory information in order to generate appropriate adaptive responses. Maintaining the balance of the body in a vertical position requires precise motor coordination, which is controlled by the CNS and manifests itself in the form of an executive corrective response with regard to the detected disturbance [9]. This process consists of stimulating the work of motor units in specific temporal and spatial relationships. Controlling posture is about providing the body with a specific posture. The involvement and influence of individual postural control systems in ontogenesis may change and can effectively compensate for the failure of one of them [10]. This study’s design was related to the search for the causes of scoliotic changes. In these changes, more and more attention is being paid to discrete neurological dysfunctions in the form of sensory information and balance deficits [11]. In the etiopathogenetic understanding, scoliotic changes are merely a symptom and an external expression of unrecognized pathology. The concept of multifactorial, including genetically determined, discrete changes in the CNS, causing disorders in the development of the spine and body posture, is gaining more and more supporters. The search for and the adaptation of new diagnostic and therapeutic methods will allow for the effective treatment of scoliotic changes [12]. Sensory integration and balance deficits negatively affect maintaining an upright body posture. They can also impair postural stability, and thus, have detrimental influence on the body posture and development of a child’s spine [13].

The aim of this study was to assess the relationship between sensory integration and balance deficits as well as scoliotic changes in young schoolgirls.

## 2. Material and Methods

The study comprised 54 girls aged 11 years with scoliotic changes. Conducting body posture examinations in girls aged 11 is related to the critical period of posturogenesis, which is the growth spurt. During the school period, attention should be paid to 2 critical periods of posturogenesis. The first critical period, at the age of 6–7, is associated with a change in the child’s lifestyle. The essence of this change lies in the transition from a free lifestyle, movement, effort, and rest, individually regulated by the child, to an imposed school system of maintaining a seated position for several hours, often in inappropriate conditions. Therefore, during this period, it is important to ensure that the child has the right living, learning, and resting conditions. The second critical period of posturogenesis is associated with the pubertal leap (girls: 11–13 years old, boys: 13–14 years old). An intensive increase in the length of the lower limbs and trunk, change in body proportions, and the current arrangement of centers of gravity, lack of simultaneous coverage of these changes with muscle strength, inadequacy of the existing feeling, and habit of posture in response to the changed morphological conditions, create a situation in which the deepening of postural defects and scoliosis is particularly frequent. This situation is exacerbated by the burden of the school curriculum, requiring many hours of sitting during lessons. In light of these remarks, the need for special care by parents, teachers, physiotherapists, and physicians becomes obvious. At the same time, it should be realized that this period often offers a final opportunity to compensate for existing deviations, as they decrease significantly after the end of growth. The end of the growth spurt, marked in girls by the first menstruation, is the moment from which the child requires special care for about a year. The growth spurt is one of the stages of the puberty process that takes place between the ages of 9 and 13. It starts suddenly, almost 2 years before puberty. Its peak usually falls a year before the first menstruation. The inclusion criteria were children of 11 years of age, with no certificate of physical and intellectual disability, with no diagnosed syndromes or congenital defects of the CNS and motor organs preventing proper psychomotor development, with no disorders that may be the cause of pathological body posture, i.e., genetic syndromes, hormonal disorders, neuromuscular diseases, and congenital defects of the locomotor system, with written consent of parents or guardians for the child to take part in the testing.

The research was conducted in 2019 at the Posturology Laboratory of Collegium Medicum, Jan Kochanowski University in Kielce. The tests were performed with the prior written consent of the respondent’s parent/guardian. All procedures were performed in accordance with the applicable 1964 Declaration of Helsinki and with the consent of the University Bioethics Committee of Jan Kochanowski University in Kielce (Resolution No. 37/2018). The Clinical Test of Sensory Integration and Balance (CTSIB) of the Biodex Balance System platform was used to analyze deficits in sensory integration and balance (Figure 1). A diagram with 4 conditions was used: eyes open—hard surface; eyes closed—hard surface; eyes open—soft surface; eyes closed—soft surface. During the eyes open hard surface test, the integration of visual, vestibular, and proprioceptive elements was examined. During the eyes closed hard surface test, integration of vestibular and proprioceptive elements was studied. During eyes open soft surface test, the integration of visual and vestibular elements was evaluated. In contrast, during the eyes closed soft surface test, the function of the vestibular elements was assessed. All children were informed about the operation of the Biodex Balance System before entering the device. During the eyes closed test, assistance from a physiotherapist in the event of loss of balance was provided and the child made 3 preliminary attempts before the test was performed. Changes in spinal and body posture were assessed using the Diers Formetric III 4D optoelectronic method (Figure 2).

This allows photogrammetric registration of the back surface using the raster stereography process. Based on the obtained data, a precise, 3-dimensional model of the surface of the back and spine is created. Taking the anatomical and biomechanical assumptions of the model into account, it was possible to calculate fixed anatomical points, spinal curvatures, and parameters of the spatial form regarding the spine and pelvis.

The Diers Formetric III 4D method is a non-contact, automatic and, above all, non-radiation method of measuring the statics of the body and spine. The device allowed for a 3-dimensional diagnosis of scoliotic changes in the spine and body posture. The study was based on the assessment of habitual posture/stance. The measurement was carried out via the Di-CAM program using the ‘Average’ test option, which consisted of taking a sequence of 12 photos. By creating an average value, variance in posture was reduced, and thus, the clinical significance of the examination was improved. The following parameters describing scoliotic changes within the spine and body posture were analyzed: spine curvature angle (°), pelvic tilt (°), deviation from the vertical line (mm), rotation of vertebrae surface (°) and lateral deviation (mm).

However, when the curvature of the spine was equal to or greater than 10°, an X-ray was ordered. Scoliosis was diagnosed when the Cobb angle was equal to or greater than 10°.

Statistical methods. Before using a parametric test, the assumption of normality was verified using the Shapiro–Wilk test. The distributions of all variables were normal. Multi-variable regression models were also applied. The homogeneity of variable variance was tested using Levene’s test. The *p*-values yielded from Levene’s test were larger than 0.05, thus, the assumption of homogeneity of variance has been not violated. The relationships between the analyzed independent variables were determined using Pearson’s correlation analysis. Correlation analysis showed very low correlations between the independent variables. In addition, the VIF analysis had a mean value of 1.2 (SD = 0.23) for all variables. All statistical analyses were performed using Statistica 12.0 (TIBCO Software Inc., Palo Alto, California, CA, USA) and Microsoft Office (Redmont, Washington, DC, USA). The level of *p* < 0.05 was adopted as being of statistical significance.

## 3. Results

Table 1 presents the most important predictors significantly related to the Clinical Test of Sensory Integration and Balance. The mean value of the angle of curvature of the spine was (x = 18.50°), the pelvic tilt was (°) (x = 3.08), the rotation of vertebrae surface (°) was (mm) (x = 6.28 mm) and the lateral deviation (mm) (x = 3.21 mm) (Table 1).

The most important predictor for the CTSIB open eyes, hard surface, was the angle of curvature of the spine (°). Regression analysis indicated that if the angle of the spine curvature (°) increased by one unit, the mean score for this test increased by 0.11 (Table 3). The most important predictor for the variable of CTSIB closed eyes, hard surface, was also the angle of curvature of the spine (°). Regression analysis indicated that if the value of the spinal curvature angle (°) increased by one unit, the mean score of this test increased by 0.02 (Table 3). The most important predictor for the variable of CTSIB open eyes, soft surface was also the angle of curvature of the spine (°). Regression analysis indicated that if the value of the spinal curvature angle (°) increased by one unit, the mean score of this test increased by 0.13 (Table 3). The most important predictor for the CTSIB closed eyes, soft surface, was at the pelvic tilt (°) (Table 3). Regression analysis indicated that if the value of the pelvic tilt (°) increased by one unit, the mean score of this test increased by 0.11 (Table 3).

In turn, the most significant predictor for the variable clinical test CTSIB open eyes, hard surface, found that the hard surface was the surface rotation of the vertebrae (°). Regression analysis showed that if the vertebral surface rotation (°) increased by one unit, the mean score for this test increased by 0.15 (Table 4). For the clinical test CTSIB closed eyes, hard surface, the most important predictor was also surface spinal rotation (°). Regression analysis suggested that if the rotation value of the spinal area (°) increased by one unit, the mean score for this test increased by 0.67 (Table 4). In contrast, the most important predictor for the CTSIB open eyes, soft surface, was lateral deviation (mm) (°). Regression analysis indicated that if the pelvic tilt (°) increased by one unit, the mean score for this test increased by 0.68 (Table 4). For the clinical test CTSIB closed eyes, soft surface, the most important predictor was also surface spinal rotation (°). Regression analysis suggested that if the rotation value of the spinal area (°) increased by one unit, the mean score for this test increased by 0.33 (Table 4).

## 4. Discussion

Although scoliotic changes are obviously distortions of the spine and posture, they are also an effect of the body’s compensatory abilities, allowing the head and shoulder girdle to remain positioned above the pelvis. The final shape of the spine is the result of deformation processes and the compensatory reactions, due to which the body maintains a general orientation of the body at the expense of disturbing its own form [14,15]. In order for the postural reflexes to run properly, it is necessary to properly integrate processed information. The SI model is based on the processing of sensory impressions in the CNS, which enables adaptation for a given situation [16]. SI is the ordering of sensory impressions in the CNS that allows it to learn and behave in a way that is appropriate for a given situation. Individual information from the sensory organs is compared and put in the right order and in the right place. The work of the CNS requires the continuous processing of stimuli. Neurons must receive stimuli for a network of connections to arise between them. It is important that the information is unambiguous and that it arrives without distortion. Improving SI determines the psychomotor and postural development of a child. For proper development, it is necessary to integrate the sensorimotor with visual and auditory perceptions as well as with the interaction of the vegetative and limbic systems [17]. This merging of information from individual sensory channels takes place right after birth and is gradually improved, thanks to which the child achieves an increasingly complex perception and has better and better adaptive responses. Sensory integration develops mainly in the first 10 years of life [18,19]. A body scheme arises in the consciousness by integrating stimuli at different levels of the CNS. Not all stimuli received by the receptors reach the cerebral cortex and not all are realized [20]. The basis for forming a body scheme comprises the lowest levels of sensory integration. The degree of excitability and concentration of attention depend on the processes of inhibition and the priming excitations in the CNS, where the reticular formation (formatio reticularis) plays a special role. Individual sensory systems can be stimulated or inhibited by the neurons of reticular formation, which recognize those that are too weak, strengthen them, or that suppress excessively intense stimuli before reaching the appropriate CNS structures (subcortical nuclei, cerebral cortex, and cerebellum). The basic feature of the CNS is the functional integration of all its parts [21].

Both the excess of sensory information and its deficiency and its inability to interpret its strength, extent and/or direction of action can result in various disorders. Disturbances in the sense of balance and in deep and superficial sensations may be caused by hypofunction, i.e., reduced insufficient sensitivity of the senses, or hyperfunction, i.e., hypersensitivity. Hypofunction, in the sense of insufficient sensitivity, is manifested by a lack of information about one’s own body and space, which, in turn, causes a movement disorder in the form of stereotypical movements [22].

In the present study, there was a significant relationship between sensory integration and balance deficits as well as the spinal curvature angle (°), vertebral surface rotation (°), pelvic tilt (°), and lateral deviation (mm). As the deficits in sensory integration and balance increase, so do scoliotic changes. The results of our own research are confirmed by those obtained in the study conducted by Simoneau et al. [23], who found that balance control depends on the availability and integrity of sensory signals, as well as the efficiency of balance control mechanisms [23]. Maciaszek et al. [24] conducted research aimed at assessing sensory integration therapy (SIT) for postural stability in children. In the study, it was shown that SIT has a positive effect among children demonstrating low levels of postural stability. Significant improvements were noted in children with a low level of postural stability, which is important not only for the current functioning of these children, but also for their future, in terms of the prevention of falls and injuries [24]. In the study carried out by Gauchard et al. [25], the authors demonstrated that dynamic and static balance affect the occurrence of increased idiopathic scoliosis diagnosis frequency, which the Clinical Tests of Sensory Integration and Balance test performed in this study confirms, with the open/closed eyes, hard surface, and the open/closed eyes. soft surface, sub-tests. In the static test, the location of the curvature, the number of arches and the size of the curvature of the spine were significant, while in the multi-scale scoliosis dynamic test, with a large angle of the curvature, the researcher noticed a deterioration in the ability to maintain balance, which was also confirmed in this study [25]. On the other hand, a group of researchers, i.e., Eshraghi et al. [26], investigated the parameters of the static and dynamic balance among 14-year-old girls with hyperphosis. They further compared these results with those for children without defects. In their research, the authors found significant differences between the groups. The mean parameters of dynamic balance were worse in girls with hyperphosis [26]. In idiopathic scoliosis and in healthy subjects, Bruyneel et al. [27] assessed postural control with open and closed eyes on the Kistler power platform. They also found an impact of the deficits in the proprioceptive-vestibular information on the development of scoliosis [27]. Research on postural stability includes tests with open and closed eyes. In another study, Pialasse et al. [28] emphasized the deterioration of body stability in the case of short-term exclusion of visual control [28]. Albertsen et al. [29] studied 96 healthy children who stood on a force plate for 60 s. They measured the effects of the support width (feet apart, FA; feet together, FT), vision (eyes open, EO; closed, EC), and cognitive load (single task, ST; double task, DT) on the anterior-posterior (AP) and ranges medial-lateral (ML) surface areas as well as the planned velocity of the centre of pressure (COP) trajectory. The results of the research conducted by the authors have shown that eyesight particularly affects stability of posture [29].

Treatment using the SI method consists of providing a variety of strong stimuli and working on recreating the image of the body and the environment. Treatment involves increasing the tension of the elimination system by stimulating the deep sensing system and the sense of balance, as well as limiting the inflow of stimuli. Minimal perinatal CNS damage (minimal brain dysfunction) may lead to not only psychomotor retardation in the first year of life, but also to muscle tension disorders, abnormal movement, and postural patterns. These damages can also affect the nerve pathways and structures, the functions of which we can only assess in later years and often only then can they be healed. Examples include postural defects and scoliotic changes [30]. The causes of SI disorders can lie in the sensory or motor systems. In the sensory system, the vestibular organ, the deep sensation system, the visual system, and the superficial sensing system are mainly responsible for integration. The aim of treatment in sensory integration disorders is normalizing the reception of tactile stimuli and regulating of muscle tone; making the movement aware by rebuilding the inner body feeling, sensing movement, and improving deep sensation; to balance regulation by improving static support, posture, and adaptive responses in maintaining balance while standing and during any action; to achieve body scheme restoration by improving visual motor skills and eye-hand-leg coordination; the improvement of visual perception in terms of shape, stability of form, and location in space; improving the perception of spatial conditions and the image of space; to achieve motor planning by teaching the direction of movement and body coordination in motor activity; the improvement of auditory self-control regarding sounds and noise and the ability to isolate sound and imitate rhythm [31].

For sensorimotor stimulation, devices are used to encourage the child to be active, providing a variety of stimuli. These are stable ladders, hanging ladders, ropes, hammocks, balancers, dry pools, slides, equally inclined, low trolleys on wheels, obstacle courses, and tunnels [32,33]. Exercises on balance platforms using the biofeedback method are also useful. This study was limited by the relatively small number of respondents (54 girls). In the future, we plan to study girls and boys comprising larger study groups and representing different age groups. The value of our research is in the use of modern, non-invasive, and objective computer methods to assess body and spina posture (Diers Formetric III 4D optoelectronic method) to determine deficits in sensory integration as well as balance (Clinical Test of Sensory Integration and Balance (CTSIB) on the Biodex Balance System platform).

## 5. Conclusions

There was a significant relationship between sensory integration and balance deficits as well as spine curvature angle (°), vertebral surface rotation (°), pelvic tilt (°), and lateral deviation (mm). As the deficits in sensory integration and balance increase, so do scoliotic changes. The integration of sensory systems has a positive effect on the structure of intended and controlled movement as well as body posture and the development of the spine. In the treatment of scoliotic changes, one should also consider exercises improving sensory integration as well as position and balance responses.

## Figures and Tables

**Figure 1 sensors-23-01172-f001:**
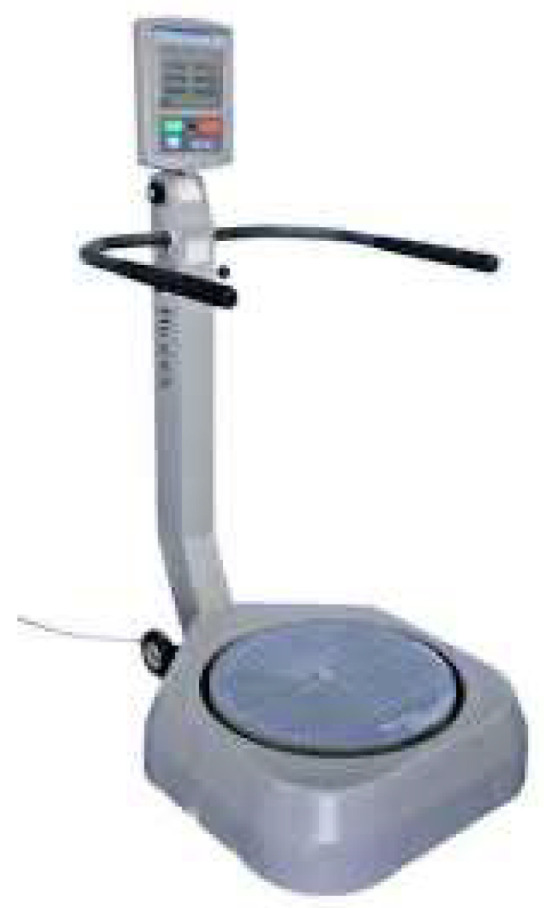
Biodex Balance System platform.

**Figure 2 sensors-23-01172-f002:**
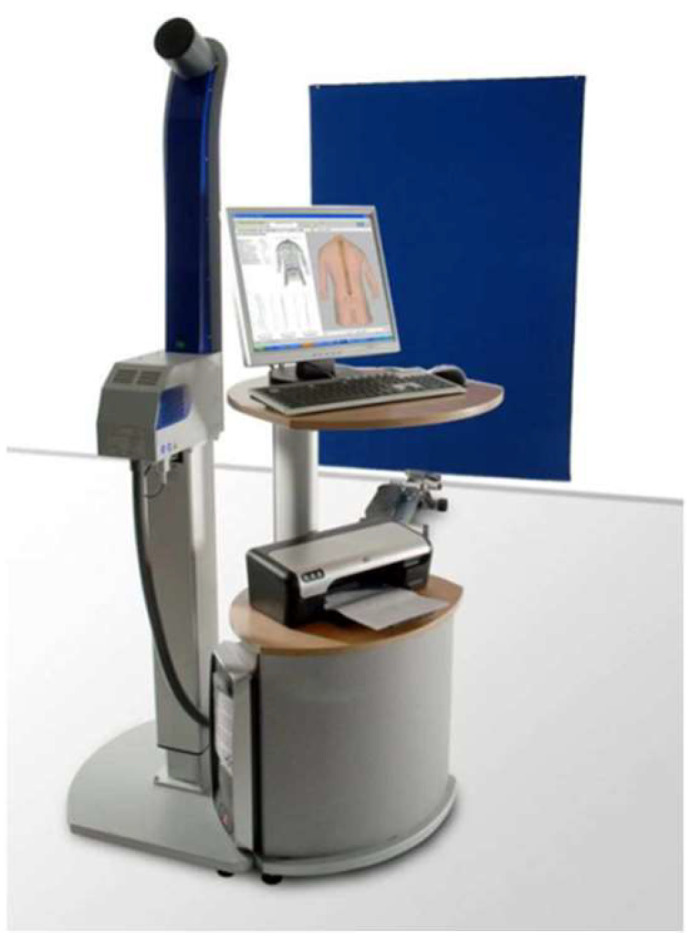
Diers Formetric III 4D.

**Table 1 sensors-23-01172-t001:** Variables of tested parameters (x: arithmetic average; s: standard deviation; V: coefficient of variation).

Variables	X	S	V
Angle of curvature of the spine (°)	18.50	10.16.	3.56
Pelvic tilt (°)	3.08	3.53	114.42
Rotation of vertebrae surface (°)	6.28	0.48	7.76
Lateral deviation (mm)	3.21	1.94	60.50

CTSIB open eyes, hard surface was (x = 1.84°); CTSIB closed eyes, hard surface was (x = 2.25°); CTSIB open eyes, soft surface was (x = 3.59°); and CTSIB closed eyes, soft surface was (x = 4.05°) (Table 2).

**Table 2 sensors-23-01172-t002:** The Clinical Test of Sensory Integration and Balance in girls with scoliosis.

Variables	X	S	V
CTSIB Open Eyes, Hard Surface	1.84	1.31	0.80
CTSIB Closed Eyes, Hard Surface	2.25	2.18	1.93
CTSIB Open Eyes, Soft Surface	3.59	2.09	0.95
CTSIB Closed Eyes, Soft Surface	4.05	2.69	1.27

**Table 3 sensors-23-01172-t003:** Regression analysis for the Clinical Test of Sensory Integration and Balance in post-tests: open and closed eyes, hard surface, and open and closed eyes, soft surface.

CTSIB Open Eyes, Hard Surface			
Variables	Beta	B	*p*
Absolute term		1.16	0.32
Angle of curvature of the spine (°)	0.95	0.11	0.01
R_2_ = 0.96			
**CTSIB Closed Eyes, Hard Surface**			
**Variables**	**Beta**	**B**	** *p* **
Absolute term		2.00	0.09
Angle of curvature of the spine (°)	0.33	0.02	0.01
R_2_ = 0.96			
**CTSIB Open Eyes, Soft Surface**			
**Variables**	**Beta**	**B**	** *p* **
Absolute term		1.35	0.14
Angle of curvature of the spine (°)	0.89	0.13	0.01
R_2_ = 0.91			
**CTSIB Closed Eyes, Soft Surface**			
**Variables**	**Beta**	**B**	** *p* **
Absolute term		0.69	0.56
Pelvic tilt (°)	0.66	0.11	0.02
R_2_ = 0.93			

**Table 4 sensors-23-01172-t004:** Results of regression analysis for the Clinical Test of Sensory Integration and Balance in sub-tests: open and closed eyes, hard surface, and open and closed eyes, soft surface.

CTSIB Open Eyes, Hard Surface			
Variables	Beta	B	*p*
Constant term		−1.01	0.48
Surface rotation of the vertebrae (°)	0.88	0.15	0.03
R_2_ = 0.93			
**CTSIB Closed Eyes, Hard Surface**			
**Variables**	**Beta**	**B**	** *p* **
Constant term		−2.22	0.39
Surface rotation of the vertebrae (°)	0.59	0.67	0.04
R_2_ = 0.93			
**CTSIB Open Eyes, Soft Surface**			
**Variables**	**Beta**	**B**	** *p* **
Constant term		0.69	0.56
Lateral deviation (mm)	0.41	0.68	0.04
R_2_ = 0.93			
**CTSIB Closed Eyes, Soft Surface**			
**Variables**	**Beta**	**B**	** *p* **
Constant term		9.39	0.08
Surface rotation of the vertebrae (°)	0.75	0.33	0.04
R_2_ = 0.92			

## Data Availability

The data used to support the findings of this study are available from the corresponding author upon request.

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
