# Peer review of "Deficits of Sensory Integration and Balance as Well as Scoliotic Changes in Young Schoolgirls"

_sensors, 2023, doi:10.3390/s23031172_

Round 1
Reviewer 1 Report
This reviewer welcomes the opportunity to review this paper. However, some questions raise some concern in relation to the article being evaluated:
1- It is important to present the complete descriptive statistics of the participants and the information collected from them, as well as mean values and standard deviation for parametric data, and median values and interquartile distance (25%-75%) for non-parametric data.
2- There is no information on the number of participants who did not meet the inclusion and exclusion criteria, or participants who dropped out.
3- Do the values collected by the 12 photos follow a normal distribution? Wouldn't it be more correct to use the median and interquartile range?
4- Data normality values were not presented for any of the collected variables.
5- The authors talk about using multivariate linear regression. But the models present only a single explanatory variable. Were other variables tested together in the regression model, and were these excluded? This is not clear. It is necessary to present and compare the complete models (with all variables) and final models (which are already in the article).
6- Confounding variables that could contribute to a better understanding of the analyzed phenomenon were also not added to the model. Models with confounding variables should compose the article.
6- There is no result associated with ANOVA. Nor what was your purpose in the study.
7- There is no analysis of the residuals of the regression models. There are also no guarantees of low exogeneity and no guarantees of absence of multicollinearity.
8- The result of the F test is not displayed, nor are the considerations that can be taken from it.
9 - The result of the Levene test is not clear.
10- The article does not make clear what its great academic contribution was compared to what had already been done by previous works, highlighting its strong points compared to what has already been done by previous studies.
11- No limitations for the study are presented. Nor are there recommendations for future work.
12- It is recommended that the keywords do not appear in the title of the article.
13- What computational tool was used to carry out the mathematical procedures?
Author Response
Reviewer Response
I am submitting the revised version of our article “Deficits of Sensory Integration and Balance as well as Scoliotic Changes in Young School-Girls” to be considered for publication in Sensors.
We would like to thank the reviewers for their time and effort put into the review of our manuscript. It was quite a challenge for us to answer some of the greatly insightful and very detailed questions. However, we have made every effort to meet this challenge. We hope that after undergoing revision and following extensive changes, the article will prove interesting and be accepted for publication. Below, please find a detailed, point-by-point description of the changes applied in the text as well as responses to comments. Once more, we are exceptionally grateful for your in-depth review of our article. Your insight and comments will definitely allow for an increase in the substantive value of the manuscript. We hope that our detailed responses and extensive changes to the text are sufficient for publication in your renowned journal. Thank you for your devoted time and effort.
Response to comments by Reviewer 1
Point 1: It is important to present the complete descriptive statistics of the participants and the information collected from them, as well as mean values and standard deviation for parametric data, and median values and interquartile distance (25%-75%) for non-parametric data.
Response 1: Tables 1 and 2 have been added and complete statistics are provided (x- arithmetic average; s- standard deviation; V - coefficient of variation).
Point 2:There is no information on the number of participants who did not meet the inclusion and exclusion criteria, or participants who dropped out.
Response 2: Short information on the number of participants who did not meet the inclusion and exclusion criteria, or participants who dropped out, was added. Twelve girls did not meet the inclusion criteria for the study.
Point 3:Do the values collected by the 12 photos follow a normal distribution? Wouldn't it be more correct to use the median and interquartile range?
Response 3: Before using a parametric test, the assumption of normality was verified using the Shapiro-Wilk test. The distributions of all variables were normal. This information was added.
Point 4:Data normality values were not presented for any of the collected variables.
Response 4: This was not necessary.
Point 5:The authors talk about using multivariate linear regression. But the models present only a single explanatory variable. Were other variables tested together in the regression model, and were these excluded? This is not clear. It is necessary to present and compare the complete models (with all variables) and final models (which are already in the article).
Response 5: The task of modelling with the use of regression models is to determine the optimal models of variables that have direct impact on the dependent variable. Each time, all analysed independent variables were tested. However, the model is created only from statistically significant variables. There is no such thing as a comet model. There is only one model and it includes statistically significant variables, the others considered irrelevant or redundant are not included in the model and were not optimised.
Point 6: There is no result associated with ANOVA. Nor what was your purpose in the study.
Response 6: There is no result associated with ANOVA. Nor what was your purpose in the study. Answer 6: Write errors from other databases. This has already been corrected.
Point 7: There is no analysis of the residuals of the regression models. There are also no guarantees of low exogeneity and no guarantees of absence of multicollinearity.
Response 7: In models with a dozen or so cases, there is never a guarantee of low homogeneity. The model is created on the basis of matrix data. Randomisation of the selection concernining research material eliminates collinearity. This was ruled out by non-collinearity analysis. Correlation analysis showed very low correlations between the independent variables. In addition, the VIF analysis had a mean value of 1.2 (SD=0.23) for all variables. This information was added.
Point 8: The result of the F test is not displayed, nor are the considerations that can be taken from it.
Response 8: The value of the F test is not displayed because ANOVA was not used.
Point 9: The result of the Levene test is not clear.
Response 9: The p-values yielded from Levene's test were larger than 0.05, thus, the assumption of homogeneity of variance has not been violated. Information has been added.
Point 10: The article does not make clear what its great academic contribution was compared to what had already been done by previous works, highlighting its strong points compared to what has already been done by previous studies.
Response 10: The study was limited by the relatively small number of respondents (54 girls). The value of our research is the use of modern, non-invasive and objective computer methods to assess body and spinal posture (Diers Formetric III 4D optoelectronic method) and to determine deficits in sensory integration as well as balance (Clinical Test of Sensory Integration and Balance (CTSIB) on the Biodex Balance System platform).
Point 11: No limitations for the study are presented. Nor are there recommendations for future work.
Response 11: The study was limited by the relatively small number of respondents (54 girls). In the future, we plan to study a larger group of girls and boys and among different age groups of children.
Point 12: It is recommended that the keywords do not appear in the title of the article.
Response 12: Keywords corrected: Neurophysiological background of scoliosis, Diers Formetric III 4D optoelectronic method, Biodex Balance System platform.
Point 13: What computational tool was used to carry out the mathematical procedures?
Response 13: All statistical analyses were performed using Statistica 12.0 (TIBCO Software Inc., Palo Alto, California, CA, USA) and Microsoft Office (Redmont, Washington, DC, USA).
Once more, we are exceptionally grateful for your in-depth review of our article. Your insight and comments will definitely allow for an increase in the substantive value of the manuscript. We hope that our detailed responses and the extensive changes to the text are sufficient for the publication of our text in your renowned journal. Thank you for your devoted time and effort.
Yours sincerely,
Jacek Wilczyński

Reviewer 2 Report
ABSTRACT: Define study design
METHOD:
-Explain why the study was done on 11-year-old girls and not at another age, such as puberty. Why the age of 11 years was an inclusion criterion?
-Unify the inclusion and exclusion criteria. If not having congenital defects of the locomotor system is an inclusion criterion, you cannot say that congenital defects of the musculoskeletal system are an exclusion criterion. That is repeating the same thing twice. Review all the criteria.
-Define study design.
- To better understand the measurement of variables provide a photograph of the Diers Formetric III 4D optoelectronic method.
If title of table 1 is: Regression analysis for the Clinical Test of Sensory Integration and Balance in post-tests: open and closed eyes hard surface and open and closed eyes - soft Surface; in this table is missing: Clinical Test of Sensory Integration and Balance, Eyes closed – Soft Surface
If title of table 2 is: Results of regression analysis for the Clinical Test of Sensory Integration and Balance in sub-tests: open and closed eyes - hard surface and open and closed eyes - soft Surface; in this table is missing:
Clinical Test of Sensory Integration and Balance, Eyes open – Soft Surface
DISCUSIÓN: The discussion is too long and a lot of information given can be omitted.
-Describes AI levels but does not relate them to the results of your study nor do they constitute an explanation or clarification of the results obtained. I think it can be summed up a lot.
-The second and third paragraphs of the discussion re-enumerate the results. This is not necessary.
-Relate the studies of the authors you expose with those obtained in your study and discuss the differences found.
-Include in the discussion section the limitations found in this study.
Author Response
Reviewer Response
I am submitting the revised version of our article “Deficits of Sensory Integration and Balance as well as Scoliotic Changes in Young School-Girls” to be considered for publication in Sensors.
We would like to thank the reviewers for their time and effort put into the review of our manuscript. It was quite a challenge for us to answer some of the greatly insightful and very detailed questions. However, we have made every effort to meet this challenge. We hope that after undergoing revision and following extensive changes, the article will prove interesting and be accepted for publication. Below, please find a detailed, point-by-point description of the changes applied in the text as well as responses to comments. Once more, we are exceptionally grateful for your in-depth review of our article. Your insight and comments will definitely allow for an increase in the substantive value of the manuscript. We hope that our detailed responses and extensive changes to the text are sufficient for publication in your renowned journal. Thank you for your devoted time and effort.
Response to comments by Reviewer 2
Point 1: ABSTRACT: Define study design
Response 1: The study design was related to the search for the causes of scoliotic changes. In these changes, more and more attention is being paid to discrete neurological dysfunctions in the form of sensory information and balance deficits. In the etiopathogenetic understanding, scoliotic changes are merely a symptom, an external expression of unrecognized pathology. The concept of multifactorial, including genetically=determined, discrete changes in the CNS, causing disorders in the development of the spine and body posture, is gaining more and more supporters. The search for and adaptation of new diagnostic and therapeutic methods will allow for effective treatment of scoliotic changes. Sensory integration and balance deficits negatively affect maintaining upright body posture. They can also impair postural stability, and thus, adversely influence body posture and development of a child's spine.
Point 2:METHOD: -Explain why the study was done on 11-year-old girls and not at another age, such as puberty. Why the age of 11 years was an inclusion criterion?.
Response2: Conducting body posture examinations in girls aged 11 is related to the critical period of posturogenesis, which is the growth spurt. During the school period, attention should be paid to 2 critical periods of posturogenesis. The first critical period, at the age of 6-7, is connected with a change in the child's lifestyle. The essence of this change lies in the transition from a free lifestyle, movement, effort and rest, individually regulated by the child, to an imposed school system of maintaining a seated position for several hours, often in inappropriate conditions. Therefore, during this period, it is important to ensure that the child has the right living, learning and resting conditions. The second critical period of posturogenesis is associated with the pubertal leap (girls: 11-13 years old, boys: 13-14 years old). An intensive increase in the length of the lower limbs and trunk, changes in body proportions and the current arrangement of centers of gravity, lack of simultaneous coverage of these changes with muscle strength, inadequacy of the existing feeling and habit of posture to the changed morphological conditions, create a situation in which the deepening of postural defects and scoliosis is particularly frequent. This situation is exacerbated by the burden of the school curriculum, requiring many hours of ‘slouching’ over lessons. In light of these remarks, the need for special care by parents, teachers, physiotherapists and physicians becomes obvious. At the same time, it should be realised that this period often offers a final opportunity to compensate for existing deviations, as they decrease significantly after the end of growth. The end of the growth spurt, marked in girls by the first menstruation, is the moment from which the child requires special care for about a year. The growth spurt is one of the stages of the puberty process that takes place between the ages of 9 and 13. It starts suddenly, almost 2 years before puberty. Its peak usually falls a year before the first menstruation.
Point 3:-Unify the inclusion and exclusion criteria. If not having congenital defects of the locomotor system is an inclusion criterion, you cannot say that congenital defects of the musculoskeletal system are an exclusion criterion. That is repeating the same thing twice. Review all the criteria.
Response3: This has been corrected.
Point 4:Define study design.
Response 4: The study design was related to the search for the causes of scoliotic changes. In these changes, more and more attention is paid to discrete neurological dysfunctions in the form of sensory information and balance deficits [11]. In the etiopathogenetic understanding, scoliotic changes are merely a symptom, an external expression of unrecognised pathology. The concept of multifactorial, including genetically=determined, discrete changes in the CNS, causing disorders in the development of the spine and body posture, is gaining more and more supporters. The search for and adaptation of new diagnostic and therapeutic methods will allow for effective treatment of scoliotic changes [12]. Sensory integration and balance deficits negatively affect maintaining upright body posture. They can also impair postural stability, and thus, adversely influencing the body posture and development of a child's spine [13]. The aim of the study was to assess the relationship between sensory integration and balance deficits as well as scoliotic changes in young school-girls.
Point 5:To better understand the measurement of variables provide a photograph of the Diers Formetric III 4D optoelectronic method.
Response 5: A photo of the Diers Formetric III 4D optoelectronic method and Biodex Balance System platform has been posted.
Point 6: If title of table 1 is: Regression analysis for the Clinical Test of Sensory Integration and Balance in post-tests: open and closed eyes hard surface and open and closed eyes - soft Surface; in this table is missing: Clinical Test of Sensory Integration and Balance, Eyes closed – Soft Surface.
Response 6: The tables have been corrected.
Point 7: DISCUSIÓN: The discussion is too long and a lot of information given can be omitted.
Response 7: The ‘Discussion’ section has been corrected.
Point 8: Describes SI levels but does not relate them to the results of your study nor do they constitute an explanation or clarification of the results obtained. I think it can be summed up a lot.
Response 8: The ‘Discussion’ section has been corrected.
Point 9: -The second and third paragraphs of the discussion re-enumerate the results. This is not necessary.
Response 9: The ‘Discussion’ section has been corrected.
Point 10: Relate the studies of the authors you expose with those obtained in your study and discuss the differences found.
Response 10: Although the results of our own research confirm the results obtained in studies conducted by other researchers, it is difficult to tell the difference. Other authors used other methodologies, studied SI disorders and scoliotic changes using other methods. The ‘Discussion’ section has been corrected.
Point 11: Include in the discussion section the limitations found in this study.
Response 11: The ‘Discussion’ section has been corrected.
Once more, we are exceptionally grateful for your in-depth view of our article. Your insight and comments will definitely allow for an increase in the substantive value of the manuscript. We hope that our detailed responses and the extensive changes to the text are sufficient for the publication of our text in your renowned journal. Thank you for your devoted time and effort.
Yours sincerely,
Jacek Wilczyński

Round 2
Reviewer 1 Report
The authors responded to all my comments satisfactorily, so that there was a significant improvement in the article.
Reviewer 2 Report
Observations and suggestuions made have been satisfactorily answered except for the study design. It must be defined that it is an observational and cross-sectional study.